# Tough Polyelectrolyte Hydrogels with Antimicrobial Property via Incorporation of Natural Multivalent Phytic Acid

**DOI:** 10.3390/polym11101721

**Published:** 2019-10-21

**Authors:** Hoang Linh Bui, Chun-Jen Huang

**Affiliations:** 1Department of Biomedical Sciences and Engineering, National Central University, Taoyuan 32023, Taiwan; hoanglinh2894@gmail.com; 2Department of Chemical and Materials Engineering, National Central University, Taoyuan 32023, Taiwan; 3R&D Center for Membrane Technology, Chung Yuan Christian University, Taoyuan 32023, Taiwan

**Keywords:** polyelectrolyte, quaternary ammonium, phytic acid, multivalent ion effect, ion bridges, specific ion effect

## Abstract

Tough and antimicrobial dual-crosslinked poly((trimethylamino)ethyl methacrylate chloride)-phytic acid hydrogel (pTMAEMA-PA) has been synthesized by adding a chemical crosslinker and docking a physical crosslinker of multivalent phytic acid into a cationic polyelectrolyte network. By increasing the loading concentration of PA, the tough hydrogel exhibits compressive stress of >1 MPa, along with high elasticity and fatigue-resistant properties. The enhanced mechanical properties of pTMAEMA-PA stem from the multivalent ion effect of PA via the formation of ion bridges within polyelectrolytes. In addition, a comparative study for a series of pTMAEMA-counterion complexes was conducted to elaborate the relationship between swelling ratio and mechanical strength. The study also revealed secondary factors, such as ion valency, ion specificity and hydrogen bond formation, holding crucial roles in tuning mechanical properties of the polyelectrolyte hydrogel. Furthermore, in bacteria attachment and disk diffusion tests, pTMAEMA-PA exhibits superior fouling resistance and antibacterial capability. The results reflect the fact that PA enables chelating strongly with divalent metal ions, hence, disrupting the outer membrane of bacteria, as well as dysfunction of organelles, DNA and protein. Overall, the work demonstrated a novel strategy for preparation of tough polyelectrolyte with antibacterial capability via docking PA to open up the potential use of PA in medical application.

## 1. Introduction

Hydrogels, a three-dimensional polymer network with high water content, has been extensively applied in biomedical applications, including tissue engineering scaffolds, tissue adhesives, drug delivery system, soft robotics, biosensors, and electronic components [1]. Recent studies have explored the potential of hydrogels in imitating human body tissue such as cartilage [2], skin [3], and muscle [4]. However, the conventional single network hydrogels with covalent crosslinks exhibited weak mechanical properties, which limits their potential uses of hydrogels in load-bearing applications [1]. The limitation stems from two major reasons: (1) ineffective energy dissipation within the polymeric network and (2) inhomogeneity distribution of polymer chain length [5,6]. To deal with these drawbacks, two outstanding approaches were introduced: nanocomposite hydrogels and interpenetrating polymer networks (IPN) [5]. Polymer chains within the nanocomposite hydrogel can be crosslinked with nanoclays through hydrogen bonds and coordinate bond to achieve structural homogeneity, thus, providing stretchable domains [6]. Whereas, in IPN, the fracture of the sacrificial first network provides effective energy dissipation and prevents catastrophic crack propagation under stress [5]. However, due to the nature of covalent bonds, damages caused by continuous stress results in permanent damage to IPN, leading to the loss of the mechanical properties over time [7]. To address the problem, many efforts have been made to incorporate or alter covalent crosslink with reversible physical crosslinker for fabricating tough hydrogels [1]. The physical crosslinkers can be temporally disrupted and recover when stress is removed, hence providing forced dissipation within the hydrated network. Owing to the dynamic behavior of physical crosslinking, hydrogels obtain extraordinary toughness along with maintaining their mechanical properties under repetitive impacts [7]. Notable physical-crosslinking strategies can include such as metal-coordination [8], hydrogen bonds [9], hydrophobic interaction [10], and ionic bonds [11]. 

Prevention of infection poses a key challenge for hydrogels to become an ideal biomaterial. While loading antibiotics into polymeric network is not an effective approach due to the wide-spread of antibiotic-resistant bacteria [12]. Alternatively, antimicrobial agents such as metal nanoparticles have been introduced and demonstrated a broad-spectrum bactericidal effect [13]. However, unrestrained release of metal nanoparticles from the nanoparticle–hydrogel composites may pose threats to human health and the environment [14]. Another alternative solution is to utilize quaternary ammonium-based (QA) polymers [15]. Polycationic polymers are able to disrupt the cytoplasmic membrane of bacteria in contact [16]. Furthermore, some studies have indicated the selective drug target of QA polymer for bacterial cells over eukaryotic cells [17,18]. Due to these attractive properties, Gan et al. have successfully interpenetrated quaternized chitosan into polydopamine (PDA)-derivatived polyampholyte via electrostatic interaction to enhance both antibacterial and mechanical properties of the hydrogels [19]. Nevertheless, contact-active strategy suffers from accumulation of dead bacteria on surfaces, which hinders the accessibility of antimicrobial properties and leads to the growth of bacteria and formation of biofilms [20]. In addition, our previous study has utilized zwitterionic polymer or polyethylene glycol to preventing the initial adhesion of bacteria on hydrogels [21,22,23]. Yet, a small amount of bacteria attachment can eventually generate biofilm formation on the antifouling surface. Thus far, antibacterial mechanism based on the “kill-and-release” strategy is proven to be the most promising approach [23]. The strategy combines both contact-active and passive-defense mechanisms into a coating surface. Hence, surface antibacterial properties of stimuli-responsive polymer can be switched reversibly under external stimulus (such as temperature [24], pH adjustment [25] and light [26]), aiming at prolonging antibacterial effect of the materials. In our previous study, we have proposed a novel “kill-and-release” strategy via docking counterions into quaternary ammonium-based poly((trimethylamino)ethyl methacrylate chloride) (pTMAEMA) brushes for reversible switching between bactericidal and bacterial-repellent functions [27]. As a result, the polyelectrolyte brushes demonstrated robust and regenerative cycles of killing and releasing bacteria by changing ion-pairing strength between positively charged surface and counterions. Nevertheless, in a highly water-absorbent hydrogel, it is problematic to employ the same approach into the polymeric network while avoiding mechanical damage to the hydrogels during operation. Therefore, it is important to employ a novel strategy to enhance antibacterial capability along with significantly improving the mechanical properties of the hydrogels. 

Dating back to ancient times, plant extracts have been considered as one of the main bio-resources for drug discovery. With the surge of antibiotic-resistant bacteria recently, more and more attention has focused on the antimicrobial capability of herbal-originated substances, including tannin, quinones, and flavonoids [28]. These secondary metabolites have not only shown substantial antimicrobial capability against a wide range of pathogenic bacteria, fungi, and yeast via growth and protease inhibition and cell wall disruption, but also provide additional benefit for the body such as anti-inflammatory, antithrombotic and antioxidant [29,30,31]. Inspired by the compelling characteristics of a plant-based protocatechualdehyde in the formation of imide and coordinative bonds, Cheng′s group has recently introduced a novel approach to fabricate a supramolecular multistimuli-responsive hydrogel with antibacterial capability [32]. Another plant derivative, phytic acid (PA) has been broadly applied in a variety of purposes, ranging from enhancing mechanical properties of materials [33,34] to therapeutic uses such as anticancer drug [35]. Yet, to our knowledge, the fabrication of an antibacterial hydrogel incorporating with PA has not been explored. 

The ionotropic gelation technique, a process facilitating the crosslinking capability of multivalent ions, has been used in the pharmaceutical field [36]. Based on the same concept, Yang et al. has successfully utilized a series of multivalent cations (Ca^2+^, Ba^2+^, Sr^2+^, Al^3+^ and Fe^3+^) to crosslink alginate/poly(acrylamide) hydrogel, thus, enhancing the mechanical properties of the polyanionic network [37]. As a result, the trivalent metal-ion/alginate hydrogel obtained high stiffness (Young′s modulus achieved over 0.1 MPa) and high tensile strength (> 900 kPa). The electrostatic interaction of pTMAEMA with counterions has been reported in polyelectrolyte brushes [27,38,39,40]. Under the influence of ionic strength, the reversible switch from an extended state in low ionic strength to a coiled state in high ionic strength [41]. The robust ionic-responsive behavior of the strong polyelectrolyte has been successfully applied to modulate the surface properties of polymeric brushes, including wettability, friction, stiffness, lubricity, and protein adsorption, in a controllable fashion after docking with proper anions [27,40,42]. It is also worth mentioning that the ionic-responsive behavior of polyelectrolyte cannot be simply clarified based on the Debye-Hückel theory, which considers ions as point charges [43]. Instead, the difference in the nature of ions, such as ion size, solubility, and charge density and ion valency, should be considered for the overall comprehension of ion-ion interaction [37,44]. Beyond the ion-pairing strength, the use of ions to tune these properties of polyelectrolyte was proposed to be governed by several contributing factors, including ion specificity, hydrogen bonding and multivalent ion effect [42].

Based on previous findings [27,37,42], we proposed a novel dual-crosslinked pTMAEMA hydrogel obtaining both high fracture strength and antimicrobial capability by promoting non-covalent interaction between quaternary ammonium headgroups of pTMAEMA and multivalent-anionic PA. The compressive mechanical test and rheological test were first conducted to evaluate the effect of PA concentration on the mechanical strength and elasticity of pTMAEMA-PA. Next, the fatigue-resistant properties of pTMAEMA-PA were revealed in the compressive loading-unloading test. The ionic crosslinking mechanism of pTMAEMA-PA was evaluated by the stability test. In addition, a comparative study of swelling ratio and mechanical properties of pTMAEMA-anion complexes was further performed to elaborate other contributing factors in the enhancement of mechanical properties of pTMAEMA hydrogels. The anions in the study include Cl^−^, SO_4_^2−^, citrate^3−^ and sodium hexametaphosphate^6−^ (PP^6−^). The physical interactions between PA and pTMAEMA were confirmed by Fourier-transform infrared spectroscopy (FTIR) spectra analysis. For the antibacterial evaluation of pTMAEMA-PA, the typical pathogenic strains of bacteria including gram-negative *Escherichia coli* (*E. coli*) and gram-positive *Staphylococcus epidermidis* (*S. epidermidis*) were used in bacterial adhesion test and antibacterial diffusion test. 

## 2. Materials and Methods 

### 2.1. Materials

2-(Methacryloyloxy) ethyltrimethylammonium chloride solution (TMAEMA) were purchased from Alfa Aesar (Ward Hill, MA, USA). Sodium sulfate (SO_4_^2−^) was purchased by Showa Chemical Industry (Tokyo, Japan). *N,N′*-Methylenebis(acrylamide) (MBAA), sodium citrate tribasic dehydrate (Citrate^3−^), sodium hexametaphosphate (PP^6−^) were purchased from Sigma Aldrich (St. Louis, MO, USA). Phytic acid (PA), 2-hydroxy-2-methylpropiophenone were purchased from Tokyo Chemical Industry Co., Ltd (Tokyo, Japan). LIVE/DEAD BacLight LIVE/DEAD BacLight was bought from Invitrogen (Carlsbad, CA, USA). Tryptic Soy broth, Luria-Bertani agar and broth were bought from Becton Dickinson (Franklin Lakes, NJ, USA) and Neogen (Lansing, MI, USA), respectively. Water used in these experiments was purified by a millipore water purification system with minimum resistivity of 18.0 MΩ·m.

### 2.2. Preparation of Polycationic Hydrogels

The polycationic pTMAEMA hydrogels were synthesized by photo-initiated radical polymerization as demonstrated in Scheme 1. A mixed aqueous solution with the concentration of 2 M of TMAEMA, 0.5 mol % crosslinker MBAA, 0.1 mol % 2-hydroxy-2-methylpropiophenone relative to the total concentration of monomer and 0.5 M NaCl was carefully prepared into a reaction flask. The mixed solution was degassed with nitrogen for 1 h. Next, 2 mL of the solution were added into one well of 24-well-plate culture plates, and the polymerization of the hydrogels were carried out under irradiating UV light (UV-LED, Brightek, Taoyuan, Taiwan) with the intensity of 100 mW/cm^2^ for 30 min. After the polymerization, the as-prepared samples were immersed into DI water for 1 day, and the water was changed three times during the time period to remove excessive residues. The final hydrogels were dried in oven and then immersed in different electrolyte solutions for the evaluation of swelling ratio and the mechanical properties of the gels. Hydrogels samples after docking couterions (Cl^−^, SO_4_^2−^, Citrate^3−^, PP^6−^ and PA) were generally termed as pTMAEMA-anion-Co, where anion is the counterion in electrolyte solution and Co is the loading concentration of counterions in mol/L.

### 2.3. Compressive Mechanical Test

Compressive mechanical properties of hydrogels samples were measured with a universal testing machine (QC-513M1F, Cometech, Taichung, Taiwan). To obtain the compressive strain-stress curves, cylindrical hydrated hydrogels of pTMAEMA-anion were compressed to their maximum strain between two parallel plates with a deformation rate of 0.5 mm/min at room temperature. Engineering fracture stress σ_f_, fracture strain ɛ_f_ were recorded. Young′s modulus (*E*) was calculated from the initial slope of the compressive stress−strain curve with strain from 5% to 10%, while maximum compressive modulus (*E*_max_) was obtained from the slope within the range of 5% strain before fracture. The compressive toughness (*T*_c_) was calculated as deformation energy under uniaxial compression per unit volume of the samples. 

For the compressive loading-unloading test, pTMAEMA-PA-1 hydrogel was first compressed to 60% and 70% deformation at the rate of 6 mm/min and then unloaded at the same deformation rate to complete one loading-unloading cycle. The successive cycle was performed 10 times with the same hydrogel. Hysteresis energy in each cycle was estimated by recording as the area within the loading−unloading curves of 10 cycles.

### 2.4. Rheological Test

Frequency Sweep test was performed with a rheometer DHR-2 (TA instrument, New Castle, DE, USA) to obtain the storage modulus (*G′*), loss modulus (*G″*) and loss factor (tan δ) of pTMAEMA-PA. The frequency sweeps were conducted with an angular frequency range of 0.1−100 rad/s and a strain of 1% at room temperature. 

### 2.5. Stability Test

To investigate the counterion exchange effect on the hydrogel, pTMAEMA-PA-1 prepared as described above were immersed in excess volume of a NaCl solution at different concentrations for 1 day to reach equilibrium. The change in volume and Young′s modulus of the hydrogels under the concentration of saline solution were analyzed to assess the stability of ionic bonds within pTMAEMA hydrogels. The volume of pTMAEMA-PA was measured before and after immersing in NaCl solution by using a vernier caliper. The volume change (% to original) was calculated as the ratio of the volume of pTMAEMA-PA-1 after ion exchange process to their original volume. The Young′s modulus (compressive modulus at 5%–10% strain) of the hydrogels was obtained in a compressive test with the deformation rate of 6 mm/min at room temperature. 

### 2.6. Swelling Ratio Measurement

The swelling ratio of hydrogels was assessed by weighting the mass of fully hydrated and dried hydrogel. The swollen cylindrical-shaped samples as described were fist weighted as *W*_s_. Then, the samples were rinsed with DI water and dried in oven for 1 day and weighted as *W*_d_. The swelling ratio of the samples was calculated by the equation as below: Swelling ratio (%) = (*W*_s_ − *W*_d_)/*W*_d_ × 100%

All data recorded were averaged from three repeated experiments. 

### 2.7. Fourier Transformed Infrared (FTIR) Spectra Analysis of pTMAEMA, PA and pTMAEMA-PA Hydrogels

The formation of ionic bonds and hydrogen bond between quaternary ammonium pendants and PA was confirmed by FT/IR spectra (FT/IR-410, Jasco, Tokyo, Japan). For sample preparation, after submerging pTMAEMA dried gels into PA solution concentration of 1 M, pTMAEMA-PA-1 hydrogels were washed for three times with DI water to remove free and loosely bound PA in the hydrogels. The samples were then dried in the oven at 80 °C for 1 day. Then, the samples were carefully grinded into fine powder and vacuumed for 2 h before the experiment. pTMAEMA hydrogels and PA were also prepared individually and measured as control samples. 

### 2.8. Bacterial Attachment Test

*Escherichia coli (E. coli)* (ATCC®25922™) were used in the experiments. After overnight culture in the incubator at 37 °C. The bacteria were collected by centrifugation at 3000 rpm for 5 min. Cell pellets were then washed with sterile PBS for three times. The bacteria solution was diluted to the optical density at 600 nm (OD600) of 0.15 using PBS, corresponding to a concentration of ∼1.2 × 10^8^ cells/mL. pTMAEMA-anion were sterilized under UV light at 254 nm for 30 min and rinsed with sterile PBS (pH 7.4) three times before immersing into 5 mL of bacterial solutions at 37 °C in 6 well-plates for 3 h at 180 rpm. The hydrogels were subsequently washed with PBS (pH 7.4) at 100 rpm for three times to discard unattached bacteria. The hydrogels were then treated with LIVE/DEAD BacLight at its working concentration for 15 min at room temperature in dark room. Fluorescent images were captured from a fluorescent microscope (Nikon Eclipse Ts2-FL, Tokyo, Japan) at 20 × magnification with excited wavelength at 488 and 561 nm and acquired emitted wavelength at green (510–540 nm) and red (620–650 nm), respectively. Quantitative data of bacterial adhesion were estimated by using ImageJ software and interpreted as the average of triplicates. 

### 2.9. Antibacterial Activity

A modified Kirby-Bauer disk diffusion method was applied to study the antibacterial activity of the hydrogels. *Staphylococcus epidermidis (S. epidermidis)* (ATCC®12228™) and *Escherichia coli (E. coli)* (ATCC®25922™) were used as Gram-positive and Gram-negative bacteria models in the test. pTMAEMA-PA hydrogels at different loading concentration of 0.1, 0.25, 0.5 and 1 M were molded into a round shape (7 mm in diameter, ~2 mm in thickness) and washed three times for 5 min with DI water. The gels were further sterile under UV light at 254 nm for 30 min. Overnight-cultured bacteria were diluted to OD600 = 0.1 (~10^8^ CFU/mL). Afterward, 10 µL of diluted gram-negative and gram-positive bacterial solutions were then plated on LB agar and TSB agar, respectively. Next, the hydrogels were then gently pressed onto cultured agar. The agar plates were then placed in an incubating oven at 37 °C and left for 10 h to assess the annular radius for each of the hydrogel. The annular radius was measured by using a vernier caliper with an error of 0.1 mm for quantitative data.

## 3. Results and Discussion

### 3.1. The Effect of Loading Concentration of PA on Mechanical Properties of pTMAEMA

The negatively-charged PA anion has been reported to form an ionic complex with positively charge amide [45], and has been known as a crosslinking agent in the synthesis of conductive polymers [46,47]. Herein, we demonstrated the effect of PA concentration on the mechanical properties of polyelectrolyte hydrogel. The pTMAEMA hydrogels were dried and then immersed into PA solutions with different concentrations (0.1, 0.25, 0.5 and 1 M). As shown in Figure 1a, the mechanical properties of pTMAEMA hydrogels enhanced with PA concentration. The fracture stress and strain of as-prepared pTMAEMA was 24.5 ± 5 kPa and 29.9% ± 4%, respectively, which are much lower than pTMAEMA-PA-1 (fracture stress at 1101.1 ± 14 kPa and fracture strain at 84.6% ± 6 %). The Young′s modulus of pTMAEMA-PA-0.1 (66 ± 3 kPa) was 4.2 times higher than that of as-prepared pTMAEMA (15.9 ± 3.7 kPa). In addition, the increase of the PA concentration did not affect the Young′s modulus (compressive modulus at 5%–10% strain offset point) of pTMAEMA hydrogels (Figure 1b). However, at near fracture point, the maximum compressive modulus of pTMAEMA-PA-1 (9.8 ± 0.47 MPa) was approximately 16.9 times higher than that of pTMAEMA-PA-0.1 (0.58 ± 0.07 MPa) (Figure 1c). We hypothesized that the changes in modulus at low and high compression reflect two mechanisms of pTMAEMA conformation within the hydrogel. As a multivalent anion, PA induces inter-chain interaction for pTMAEMA via ion-bridging effect [42]. Due to the strong electrostatic interaction of multivalent counterion, the polyelectrolyte gel becomes more compact. Additional anions further infuse into the polymer network to form intra-chain bridges and lead to more chain contraction [48]. As a result, the presence of PA enhances crosslinking density within the hydrophilic network. 

The viscoelastic behavior of pTMAEMA-PA was explored in the rheological study. For all pTMAEMA-PA samples, the storage modulus *G′* was greater than the loss modulus *G′′* and tan δ < 1, demonstrating a gel-like property of pTMAEMA-PA (Figure 2a,b). Furthermore, the increase in *G′* according with the PA concentration is due to a higher crosslinking density as well as a higher order of ion bridges distributing within pTMAEMA-PA [49]. Additionally, the tan δ is closer to 0 when the concentration of PA was 1 M, indicating the distinguish elasticity of pTMAEMA-PA-1 as the ideal elastic material obtains tan δ = 0. Overall, the result confirmed the importance of ionic interaction in the enhanced elasticity of the hydrogel, which is in agreement with the previous report [50].

### 3.2. The Fatigue-Resistant Properties and Self-Recovery of pTMAEMA-PA

The fatigue resistant properties of pTMAEMA-PA-1 were further evaluated by the cyclic compressive test. A loading-unloading cycle of the hydrogel was illustrated in Figure 3a. The areas of hysteresis loops in one loading–unloading cycle represent the energy dissipated per unit gel volume. An increase in the hysteresis area appeared as the strain increased from 60% to 70% as shown in Figure 3b. The results reflected the energy dissipation capability of physically crosslinked hydrogels under deformation [11]. Furthermore, the highly overlapping hysteresis loops in the 1st, 5th and 10th cycle indicate excellent fatigue-resistant ability of pTMAEMA-PA-1. Hysteresis energies remain relatively stable after 10 successive loading-unloading cycles (Figure 3c and Appendix A). Under repetitive stresses, the physical bonds repeatedly disassembled and reformed, leading to the energy dissipation within the hydrogel network, while protecting covalent bonds from permanent breakage. The results from the cyclic compressive test further confirmed the elasticity of pTMAEMA-PA-1, as demonstrated in the rheological study. 

### 3.3. Counterion Exchange in pTMAEMA-PA

The stability of ionic complex within pTMAEMA-PA hydrogel can be affected by ionic strength of the NaCl solution [11]. The increase in the ionic strength of saline solution led to the reduction of the Young′s modulus of pTMAEMA-PA-1, from 53.3 ± 3 to 25.9 ± 5 kPa after changing the solution of pure water to 1 M NaCl, respectively (Figure 4). Furthermore, NaCl concentration influenced the swelling behavior of pTMAEMA-PA-1. Under low ionic strength condition (< 0.15 M NaCl), pTMAEMA-PA-1 deswelled, while re-swelling was observed under a high ionic strength condition, from 0.15 to 1 M NaCl (Figure 4).

The ionic interaction between multivalent ions and polyelectrolyte is governed by two interactions: ion bridges and charge-image charge interaction [42]. The attraction of two polyelectrolytes has been widely reported due to the ion-bridging effect at a low concentration of multivalent counterions, leading to the de-solvation of polyelectrolytes [42]. However, at a high concentration of multivalent ions, the counterions can form strongly correlated liquid layer with polyelectrolyte. This further promotes surface polarization under the liquid layer and the counterions become image charges with opposite charged signs [51]. The repulsive force between incoming counterions and absorbed counterions created correlation holes, which promotes further counterion condensation within polyelectrolytes. The phenomenon was called “correlation attraction” [52]. Eventually, the repulsive forces between multivalent counterions are dominated by the correlation attraction, leading to the re-solvation and re-swelling of polyelectrolyte due to the charge inversion effect [51,52]. It is also worth mentioning that the coexistence between neutral complexes (or ion bridges) and condensation phases via correlation attraction can be achieved under some certain conditions [52]. Assuming that 1M concentration of PA is high enough for ion bridges and correlation attraction occurs concurrently within pTMAEMA hydrogel. The changes in volume and Young′s modulus in pTMAEMA-PA-1 under ion exchange effect can be understood through the theory of reentrant condensation as demonstrated in Scheme 2 [52]: In DI water, no significant change in volume and Young′s modulus of pTMAEMA-PA-1 was observed due to the strong correlation attraction and ion bridges within pTMAEMA-PA. In a low ionic strength solution (pure water to 0.15 M NaCl), monovalent Cl^−^ can displace the absorbed PA molecules near the solid/liquid interface, resulting in the de-swelling. As a result, the hydrogel shrank due to the removal of condensed counterions. Further increasing NaCl concentration led to the disassembly of ion bridges within the hydrogel. Eventually, pTMAEMA-PA-1 rehydrated and re-swelled, leading to complete charge screening and the marked increase in the volume of pTMAEMA-PA-1. 

It is worthwhile to acknowledge that the Young’s modulus represents the resistance of the material against elastic deformation (non-permanent deformation) when a stress is applied on a material. The reduction in the stiffness of pTMAEMA-PA hydrogels from the stability test confirmed the reversibility of the ionic interaction between PA and quaternary ammonium pendants. Furthermore, it is likely that two coexisting states contribute to the enhancement of mechanical properties of pTMAEMA-PA-1. After rinsing pTMAEMA-PA-1 hydrogels with DI water, the fracture stress of pTMAEMA-PA-1 was still maintained approximately at 1 MPa, indicating strong ionic interaction between pTMAEMA and PA (Appendix A). 

### 3.4. Counterion Effects on the Swelling Ratio and Mechanical Properties of Polyelectrolyte Hydrogels

The effect of the ion-pairing strength on the mechanical properties of pTMAEMA hydrogels has been demonstrated above. However, other factors, such as ion size, ion valency and ion specificity, also showed contribution in changing swelling and mechanical properties of polyelectrolyte in simulations and experiments [37,42,53,54]. Herein, we conducted a comparative study to elaborate the possible contributing factors that enhance the mechanical properties of pTMAEMA. The hydrogels were dried and then immersed in a series of counterion solutions with different ion valences (Cl^−^, SO_4_^2−^, Citrate^3−^, PP^6−^ and PA), and all solutions were kept at the same concentration. 

We first compared the swelling ratio of the pTMAEMA hydrogels after introducing different counterions (Figure 5a,b). pTMAEMA hydrogels in DI water obtained high swelling ratio of ~8000%. In addition, the swelling ratio of hydrogels decreased proportionally with the increase in ion valences of counterions, following the order of Cl^−^ > SO_4_^2−^ > citrate^3−^ > PP^6−^ > PA, which is in agreement with our previous study [27]. Additionally, the driving force at the beginning of the swelling process is based on osmotic pressure difference between the dried hydrogel and the solution [55]. In DI water, pTMAEMA hydrogels exhibit intrinsic electrostatic repulsion between neighbouring polymer chains and the excluded volume effects of the solvated chain side groups, leading to extended conformation of polyelectrolyte and the high swelling ratio of the hydrogel [40]. With the addition of counterions, the charge screening effect occurs, leading to the reduction of charge repulsion effect and the polyelectrolytes collapse [40]. Interestingly, pTMAEMA-PA obtained significantly lower swelling ratio than that of pTMAEMA-PP^6−^ regardless of their similarity in term of ion valency and chemical structure. The results suggest that certain factors can participate in pTMAEMA-anion systems beyond the ion valency. 

Therefore, pTMAEMA hydrogels were prepared by immerging pTMAEMA hydrogels in 1M electrolyte solutions to enhance the polymer collapse within the hydrogel networks. As shown in Figure 1 and Figure 6(a,b), as-prepared pTMAEMA hydrogels exhibited low fracture stress, strain and low Young′s modulus. The possible reasons behind this mechanical weakness are the low internal friction and the ineffective stress dissipation within polyelectrolyte hydrogel [5,11]. In addition, the inhomogeneous distribution of polyelectrolyte chain length between crosslinking points during the synthesis of the conventional single-network hydrogel also causes uneven force distribution within the hydrogel network [5]. As a result, the short polymer chains under low compression are likely to be damaged first, eventually leading to the complete failure of the hydrophilic network under compression. In contrast, with the addition of counterions, pTMAEMA chains collapses, resulting in an increase in the density of polymer chains in accordance with the decrease in the swelling ratio [54]. This promotes the internal friction between polymer chains, thereby, enhancing energy dissipation in the hydrogels. Comparing between pTMAEMA-anion systems (Figure 6c), the fracture stress σ_f_ displayed an order as follow: SO_4_^2^^−^ < Cl^−^ < citrate^3^^−^ < PP^6^^−^ < PA. The compressive toughness Tc recorded from each compressive test further confirmed the order (Figure 6c). The fracture stress of pTMAEMA-PA was 13 and 45 times higher than that of pTMAEMA-PP^6^^−^ and as-prepared pTMAEMA, respectively. In term of Young′s modulus, pTMAEMA-PP^6^^−^ and pTMAEMA-PA exhibit higher stiffness (50.0 ± 7.6 and 67.0 ± 9.2 kPa, respectively) compared with other pTMAEMA-anion complexes. 

The differences in mechanical properties of the pTMAEMA-anion systems can be explained as followed: 

(1) The fracture stress of pTMAEMA is influenced by ion valency. The mechanical strength of these polyelectrolyte-anion complexes was inversely proportional to the swelling ratio in the case of Cl^−^, Citrate^3^^−^, PP^6^^−^ and PA. With the increase in ion valency, the turbidity can be observed in pTMAEMA-PP^6^^−^ and pTMAEMA-PA occurs likely due to the de-solvation of pTMAEMA by strong ionic coupling of PP^6^^−^ and PA with quaternary ammonium pendants [42,56]. The results further indicate the multivalent ion effect on the polyelectrolyte in the formation of inter- and intra- ion bridges. 

(2) pTMAEMA-Cl^−^ showed a higher swelling ratio than that of pTMAEMA-SO_4_^2^^−^, but displayed significantly higher fracture strength. The weak mechanical strength of pTMAEMA-SO_4_^2^^−^ can be understood by the weak ionic coupling between SO_4_^2^^−^ and quaternary ammonium pendants. Following the order of Hofmeister series, the kosmotropic SO_4_^2^^−^ is strongly hydrated, whereas chaotropic Cl^−^ is less hydrated [42,57]. It has been revealed that strongly hydrated species do not easily shed their innermost hydration shell, and weakly interact with charged pendants via competition of water molecules [57]. On the contrary, chaotropes exhibit direct ion-pairing with polyelectrolyte, resulting in the “hydrophobic collapse” [41]. 

### 3.5. FTIR Analysis

Through the compressive mechanical test, one remaining question is why pTMAEMA-PA exhibited higher mechanical strength than pTMAEMA-PP^6^^−^. To address the question as well as verifying the ionic interaction of PA and quaternary ammonium, FTIR spectra measurement was applied. Figure 7a displays the characteristic bands of pTMAEMA center at wavelength number of 3019, 2919 and 953 cm^−1^, corresponding to the asymmetric stretching mode of CH_3_-N^+^, antisymmetric stretching mode of CH_2_, and stretching mode of N^+^-(CH_3_)_3_, respectively [41]. In pTMAEMA-PA spectra, significant red shift was observed (3019 to 2957 cm^−1^) (Figure 7a). The shift in the peak position indicates ionic interaction between anions and pTMAEMA chains due to the sensitivity of the CH_2_ stretching modes [41]. The redshift observed in our study may suggest the distinct ionic interaction of kosmotropic HPO_4_^2^^−^ anions (anionic groups of PA) on polyelectrolyte chains. It is worth mentioning that the electrostatic interaction of kosmotropes is based on the indirect specific competition for water molecules between counterions and quaternary ammonium moieties of pTMAEMA [42,58]. Therefore, the interaction of kosmostropic anion may promote conformational order in polymer sidechains. In addition, the characteristic absorption peak of P–O–C at 1064 cm^−1^ appears in pTMAEMA-PA spectra, indicating the success of docking PA on pTMAEMA polymer chains as reported in literature [59]. In Figure 7b, the change in the peak position of OH bands (3445 to 3401 cm^−1^) between as-prepared pTMAEMA and pTMAEMA-PA confirmed the formation of hydrogen-bonding within pTMAEMA-PA hydrogels [34]. Furthermore, no peak shift in OH band was observed in the spectra of pTMAEMA-PP^6^^−^. From the results, we proposed hydrogen bonding formation between hydroxyl groups of PA (hydrogen bond donor) and carbonyl groups on pTMAEMA side chains (hydrogen bond acceptor) [42], which further strengthens the ionic complexation within pTMAEMA-PA hydrogels.

Overall, the proposed mechanisms behind the enhancement in mechanical properties of pTMAEMA hydrogels using PA were summarized in Scheme 1.

### 3.6. Bacterial Attachment Test of pTMAEMA-Counterions

Quaternary ammonium groups have shown their effective antibacterial capability via contact-killing mechanism [27]. However, the accumulation of dead bacterial debris attached on the surface of pTMAEMA is able to condition the surface of the hydrogels, leading to the biofilm formation [23]. Herein, bacterial attachment test was performed to investigate the interfacial properties of pTMAEMA-anion systems. As shown in Figure 8a, the reduction of bacterial attachment was significant on pTMAEMA-PP^6^^−^ and pTMAEMA-PA. Quantitative data in Figure 8b show that the levels of the dead bacterial accumulation on pTMAEMA-Cl^−^, pTMAEMA-SO_4_^2^^−^ and pTMAEMA-citrate^3^^−^ were reduced by 70%, 67% and 72%, respectively, in comparison with that on as-prepared pTMAEMA. No significant difference was found in the numbers of live bacteria on these hydrogels. However, the numbers of live bacteria on pTMAEMA-PP^6^^−^ and pTMAEMA-PA were only 9% and 1% in comparison with that of as-prepared pTMAEMA. As a result, the small number of bacterial attachment observed on pTMAEMA-PP^6^^−^ may be contributed by two factors: (1) PP^6−^ anions condition on pTMAEMA surface, leading to the generation of net negative charge and induce repulsive force to bacterial cells [27]; (2) osmotic pressure difference between outside and within pTMAEMA-PP^6-^ may attract water molecules on the hydrogel surface to form water layer that prevents biofouling and bacterial growth on hydrogels.

### 3.7. Antibacterial Capability of pTMAEMA-PA

The antibacterial capability of counterions was evaluated in agar diffusion test. As illustrated in Figure 9a, only pTMAEMA-PA showed an inhibition zone to suppress the growth of *E. coli*. The results confirmed the antibacterial activity of PA. In addition, no inhibition zone was shown in agar diffusion test for pTMAEMA-PP^6^^−^ even at 1 M loading concentration. Furthermore, higher loading concentration of PA led to increase in the annular radius in both gram-negative *E. coli* and gram-positive *S. epidermidis* (Figure 9b,c). The results suggest that the antimicrobial capability of pTMAEMA-PA is proportional to the loading concentration, which was verified by the PA releasing profile in Appendix A. To confirm that the antibacterial activity of PA was not entirely based on the effect of protons in acidic solution, pTMAEMA after immersing in NaCl solution (at 0.1, 0.25, 0.5 and 1 mol/L of concentration, pH adjustment to 2 with HCl) was used in as control samples. As shown in Figure 9d, no inhibition zone was observed in the agar diffusion test, indicating no effect of protons on bacterial growth in disc diffusion test.

Antibacterial capability of pTMAEMA-PA hydrogels can be described based on its chelating effect. Under neutral pH of bacterial solution, active phosphate groups of PA form chelating complex with divalent cations metal ions (Ca^2+^, Mg^2+^). The lack of these metal ions leads to the disruption of the membrane of bacteria [60]. Moreover, charged ions such as protons and other ions in solution can easily get into bacterial cells and cause further internal damage to bacterial organelles and DNA [61]. The chelating effect of PA may also cause damages to bacteria via protein dysfunction [62]. 

It is expected that PA, as a natural chelating agent, can obtain less cytotoxicity than other agents such as ethylenediamine tetraacetic acid (EDTA). The result in Appendix A further confirm a good biocompatibility of as-prepared pTMAEMA and pTMAEMA-PA-0.1 to NIH/3T3 fibroblasts after co-culturing with 4-day-prepared extraction media from the hydrogels. Some reports also mentioned the effect of PA in enhanced cell activity, cell growth and proliferation at certain doses of PA, which is in agreement with the high cell viability level of NIH/3T3 fibroblasts when exposing with the 1-day-prepared extraction medium of pTMAEMA-PA-0.1 [33,59]. In addition, the safety in PA has been demonstrated in mouse model in several studies [63,64,65]. Moreover, PA has been acknowledged by the U.S. Food and Drug Administration as a food supplement in the Generally Recognized as Safe Notification (GRAS) category [66], which highlighted the beneficial effect of PA. Quaternary ammonium pedants under ion exchange and the change in temperature may disassemble weak ionic bonds formed with anions, hence, maintaining the sustainable release of antibacterial agents from the hydrogels. Nevertheless, the loading concentration of PA can be adjusted to prevent cytotoxicity of pTMAEMA-PA hydrogels, while holding good mechanical properties and antibacterial capability. 

## 4. Conclusions

In summary, mechanical properties of pTMAEMA hydrogels can be improved by docking multivalent natural counterions. The distinct mechanical properties of different pTMAEMA-anion complexes can be explained by several factors, including ionic strength, multivalent ion effect, ion specificity and hydrogen bonding. Remarkably, pTMAEMA-PA hydrogels obtained superior fracture stress that was 13 and 45 times higher than that of pTMAEMA-PP^6^^−^ and as-prepared pTMAEMA gels. The evaluation of counterion exchange effect with pTMAEMA-PA-1 revealed that the combination of ion bridges and correlation attraction within the hydrogel contribute to the high mechanical properties of the hydrogels. Furthermore, pTMAEMA-PA-1 hydrogels displayed excellent elasticity with fatigue-resistant property, which was demonstrated in rheological and compressive loading-unloading test. Moreover, from the bacterial adhesion test and the agar diffusion test, pTMAEMA-PA hydrogels showed excellent antifouling and bactericidal properties. Overall, the ionotropic complexation between polyelectrolyte and PA offers great potential for the engineering both tough and antimicrobial materials, as well as other biomedical applications in the future.

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
