# Peer review of "Tough Polyelectrolyte Hydrogels with Antimicrobial Property via Incorporation of Natural Multivalent Phytic Acid"

_polymers, 2019, doi:10.3390/polym11101721_

Round 1

Reviewer 1 Report

This work synthesized a kind of dual-crosslinked poly(trimethylamino )ethyl methacrylate chloride)-phytic acid hydrogel which is tough and antimicrobial. To verify this property, some mechanical tests and antibacterial tests had been performed. I think this paper can be accepted after fully addressing several major concerns below:

The curves of 60% 1stcycle and 70% 1stcycle are not demonstrate clearly in Figure 3b, maybe you need to change colors of the curves or zoom in on the overlap. And four successive loading-unloading cycles may not enough to show the excellent fatigue-resistant properties, five cycles or more would be better. In section 3.7, why the cell viability of pTMAEMA-PA-0.1 in day 1 can reach to 170%, and drop rapidly in day 4? It is better to show the cytotoxicity data of pTMAEMA-PA-1 which has best mechanical and antibacterial properties. Some minorerrors can be found in section 3.7 line four and label of figure 9. Need to carefully revise all of them before the submission. The authors could add the following references which would again increase the interest to general hydrogel and antibacterial material readers: Journal of Controlled Release 2018,273, 160-179; Chem. Mater.2019,31,7678-7685; Chem 2017, 3, 390−410; Adv. Mater. 2014, 26, 85−124.

Author Response

Reviewer 1

The curves of 60% 1 st cycle and 70% 1 st cycle are not demonstrate clearly in Figure 3b, maybe you need to change colors of the curves or zoom in on the overlap. And four successive loading-unloading cycles may not enough to show the excellent fatigue-resistant properties, five cycles or more would be better.

We appreciate the reviewer’s suggestion. Figure 3b and c have been replaced with the compressive loading-unloading test with 10 successive cycles. The discussion has been changed accordingly in Section 3.2. The test has been conducted according to the mentioned method in Section 2.3. In detail, pTMAEMA-PA-1 hydrogel was first compressed to 60% and 70% deformation at the rate of 6 mm/min, and then unloaded at the same deformation rate to complete one loading-unloading cycle. The successive cycle was performed for 10 times with the same hydrogel. Hysteresis energy in each cycle were estimated by recording as the area within the loading−unloading curves of 10 cycles. Figure S1 in supporting information has been added to display 10 compressive loading-unloading curves for the verification of the overlapping loading-unloading curves and the durability of the hydrogels.

In section 3.7, why the cell viability of pTMAEMA-PA-0.1 in day 1 can reach to 170%, and drop rapidly in day 4?

The cytotoxicity assay has been performed as demonstrated in Supporting Information. In particular, pTMAEMA and pTMAEMA-PA-0.1 hydrogels were cut into circular discs (7 mm in diameter), and rinsed in excess PBS solution for one day prior to the experiment. Then, the hydrogels were sterilized by immersing in 75 % v/v ethanol, and again rinsed with sterilized PBS for three times before testing. The extraction medium was prepared by immersing an individual hydrogel in 24-well plates that contained 2 mL of serum-free DMEM medium for 1 and 4 days. NIH/3T3 (mouse fibroblast) cells were cultured in DMEM media (comprised of 10% fetal bovine serum [FBS] and 1% penicillin). NIH/3T3 cells were seeded into 24-well plates at a density of 1 × 104 per well in 1 mL of media for 24 h. The medium was removed and replaced with a serum-free media overnight, then replaced by 500 μL of the prepared extraction media of hydrogels, and the cells were again incubated for 24 h. The percentage of cell viability was determined using MTT assays. MTT solution (5 mg mL-1 in PBS) was diluted by 10 times with serum-free DMEM, and added to each well. Samples were incubated at 37oC for 3 h. After that, the MTT solution was removed and the precipitated violet crystals were dissolved in 500 μL of DMSO. The absorbance at 550 nm was measured using the ELISA reader (Synergy 2, BioTek, USA). Results were interpreted as an average value from four samples.

It is important to mention that the number of cells within each well of 24-well plate was identical and only the extraction taken from pTMAEMA-PA-0.1 hydrogels after 1 and 4 days was done in the same sample. After obtaining the data, we still got the results as shown in Figure S4. The reason behind the high viability of 170% in day 1 with sample of pTMAEMA-PA-0.1 and drop rapidly in day 4 is not clear. However, the similar results were found in literature [1-2]. In order to clarify the data, we add a discussion in Page 14, Section 3.7, Line 481.

It is better to show the cytotoxicity data of pTMAEMA-PA-1 which has best mechanical and antibacterial properties.

In the cytotoxicity test, we verified the samples of pTMAEMA hydrogels and pTMAEMA-PA-0.1 complex. The data showed promising biocompatibility of the pTMAEMA-PA-0.1 with extremely low cytotoxicity after incubating with the 4-day-prepared extraction. In addition, PA has been recognized as a food additive in the Generally Recognized as Safe Notification (GRAS) inventory published by the U.S. Food and Drug Administration [3], which highlighted the beneficial effect of PA. We believed that PA is a natural product with high biocompatibility. The new discussion has been added in Page 14, Section 3.7, Line 484 with additional references.

Some minorerrors can be found in section 3.7 line four and label of figure 9.

The errors in Sec 3.7 line 4 and the label of Figure 9 have been fixed as the reviewer’s suggestion.

The authors could add the following references which would again increase the interest to general hydrogel and antibacterial material readers: Journal of Controlled Release 2018,273, 160-179; Chem. Mater.2019,31,7678-7685; Chem 2017, 3, 390−410; Adv. Mater. 2014, 26, 85−124.

We appreciate the suggestion from the reviewer. After reading through all of the references, we found that the references are a very compelling example and informative for the readers to realize the importance of natural additive in the modification of hydrogels. The brief description has been added in Page 2, Section 1, Line 94.

Reference

Meininger, S.; Blum, C.; Schamel, M.; Barralet, J. E.; Ignatius, A.; Gbureck, U. Phytic acid as alternative setting retarder enhanced biological performance of dicalcium phosphate cement in vitro. Sci. Rep. 2017, 7, 558. Wang, X.; Wen, K.; Yang, X.; Li, L.; Yu, X. Biocompatibility and anti-calcification of a biological artery immobilized with naturally-occurring phytic acid as the crosslinking agent. J. Mater. Chem. B 2017, 5, 8115-8124. GRAS Notices. Available online: https://www.accessdata.fda.gov/scripts/fdcc/?set=GRASNotices&id=381 (accessed on October 13 2019).

Reviewer 2 Report

Hoang Linh Bui and Chun-Jen Huang present an interesting study of pTMAEMA hydrogels which shwo enhanced toughness upon addition of phytic acid. This is an interesting system which in addition has anti-bacterial properties. Hence, I think the work should be published after revision.

However, I think the clarity of presentation would be improved if all resutlts related to the other anions would be moved to the supplementary material. In its preset form the manuscript is rather hard to read.

In addition the English should be polished a bit. There many typos and small errors in grammar in the manuscript.

Author Response

Reviewer:2

I think the clarity of presentation would be improved if all resutlts related to the other anions would be moved to the supplementary material. In its preset form the manuscript is rather hard to read.

Figure 6 and the following discussion in Section 3.4 are important to clarify the factors which contribute to the mechanical enhancement of pTMAEMA (multivalent ion effect, ion valency, specific ion effect and hydrogen bonding in the Section 3.5). In order to improve the clarity of presentation, we have changed Scheme 2 to Page 4 to describe the method and general findings for the easier comprehension.

In addition the English should be polished a bit. There many typos and small errors in grammar in the manuscript.

The article has been edited carefully to polish the content.

Reviewer 3 Report

The paper: “Tough Polyelectrolyte Hydrogels with Antimicrobial  Property via Incorporation of Natural Multivalent  Phytic Acid” by Bui and Huang provides very interesting results on improvement of mechanical properties of hydrogels by docking them with multivalent natural counterions.

Some English errors have been detected, such as: Line 55 instead of “includes” should be “include”; however, the paper is well written and very easy to understand. The authors provided excellence discussion of results supported with literature.

In my opinion, before the paper will be publish, the authors should move the Scheme 2 to the part in which they explain the aim of the research. This figure includes reagent structures and a global vision of the procedure.

It is a very good paper and I strongly recommend its publication.

Author Response

Reviewer:3

Some English errors have been detected, such as: Line 55 instead of “includes” should be “include”;

We appreciate the comment from the reviewer. The error in Line 55 has been fixed accordingly to the reviewer’s comment.

The authors should move the Scheme 2 to the part in which they explain the aim of the research.

We appreciate the comment from the reviewer. Scheme 2 has been moved to Page 4 to describe the method and general findings.

Editor comments: One reviewer ask for one experiment more to increase the trust and statistical results I agree with this.

We have conducted again the compressive loading-unloading test for 10 successive loading-unloading cycles to emphasize the fatigue-resistant properties of pTMAEMA-PA-1 hydrogels. The results in Figure 3b and 3c have been replaced. Figure S1 has been added to displayed the overlapping of 10 loading-unloading cycles.

Round 2

Reviewer 1 Report

I think the current revised version seems Ok for me